# Mechanism and Prospect of Gastrodin in Osteoporosis, Bone Regeneration, and Osseointegration

**DOI:** 10.3390/ph15111432

**Published:** 2022-11-18

**Authors:** Yi Li, Fenglan Li

**Affiliations:** Department of Prosthodontics, Shanxi Provincial People’s Hospital, Shanxi Medical University, Taiyuan 030000, China

**Keywords:** gastrodin, osteoporosis, bone regeneration, osseointegration, actin filament, renin-angiotensin system, ferroptosis

## Abstract

Gastrodin, a traditional Chinese medicine ingredient, is widely used to treat vascular and neurological diseases. However, recently, an increasing number of studies have shown that gastrodin has anti-osteoporosis effects, and its mechanisms of action include its antioxidant effect, anti-inflammatory effect, and anti-apoptotic effect. In addition, gastrodin has many unique advantages in promoting bone healing in tissue engineering, such as inducing high hydrophilicity in the material surface, its anti-inflammatory effect, and pro-vascular regeneration. Therefore, this paper summarized the effects and mechanisms of gastrodin on osteoporosis and bone regeneration in the current research. Here we propose an assumption that the use of gastrodin in the surface loading of oral implants may greatly promote the osseointegration of implants and increase the success rate of implants. In addition, we speculated on the potential mechanisms of gastrodin against osteoporosis, by affecting actin filament polymerization, renin–angiotensin system (RAS) and ferroptosis, and proposed that the potential combination of gastrodin with Mg^2+^, angiotensin type 2 receptor blockers or artemisinin may greatly inhibit osteoporosis. The purpose of this review is to provide a reference for more in-depth research and application of gastrodin in the treatment of osteoporosis and implant osseointegration in the future.

## 1. Introduction

Osteoporosis (OP) is a systemic bone disease characterized by reduced bone mass, damage to bone tissue microarchitecture, increased bone fragility, and susceptibility to fractures [1]. According to the *International Osteoporosis Foundation* [2], more than 8.9 million fractures caused by OP are reported each year. In China, an epidemiological survey showed that the prevalence of OP was 19.2% in people aged > 50 years and up to 32.0% in those aged > 65 years [3]. The occurrence of OP is associated with decreased sex hormones, systemic inflammation, oxidative stress, glycosylation, decreased calcium absorption, and decreased physical activity [1,4]. Currently, OP is mostly treated with bisphosphonates, calcitonin, and selective estrogen receptor modulators; however, these drugs mainly prevent further bone loss without any significant effect on osteogenesis promotion and have several adverse effects [5,6,7].

Chinese herbal medicines have the advantages of multi-site action, multi-targeted action, high safety, large dosing space, less toxic side effects, and systemic conditioning [8]. Gastrodin, epimedium, curcumin, and other herbal active ingredients have received attention, and their antioxidant, anti-inflammatory, and osteogenic properties have been proven in numerous studies [9,10,11,12]. Compared to other herbal components such as epimedium and curcumin, gastrodin has a relatively simple chemical structure and comprises polyhydroxy functional groups, which are more conducive to chemical polymerization [13,14,15]. Its pharmacokinetic characteristics and wide biodistribution facilitate sustained drug delivery. Moreover, its properties of increasing the hydrophilicity of the material surface and pro-vascular regeneration are favorable for bone healing around biomaterials [13,16,17].

However, there currently is no summary of the effects and mechanisms of action of gastrodin on bone, bone regeneration, and osseointegration. This study discusses the effects of gastrodin on OP, tissue engineering, and implant osseointegration, as well as its possible mechanisms and conjecture regarding the active use of gastrodin in the surface loading of oral implants. Furthermore, we reviewed the pharmacokinetics, toxicity, and physicochemical properties of gastrodin. Moreover, we also hypothesized the potential mechanisms of gastrodin on actin filament polymerization, renin–angiotensin system (RAS) and anti-ferroptosis in bone metabolism, and proposed that the combinations of gastrodin with Mg^2+^, angiotensin type 2 receptor blockers or artemisinin may have greater potential to anti-osteoporosis.

## 2. Oxidative Stress in Menopausal Women, Elderly Patients, and Diabetes Patients with Osteoporosis

OP is characterized by very poor bone quality due to the loss of bone mineral density (BMD), microarchitecture, and mechanical strength. It is a major cause of skeletal fragility and dysfunction in postmenopausal women and elderly patients [18]. Approximately one-third of postmenopausal women aged > 50 years are affected by OP. As the population ages, the incidence of osteoporotic fractures increases [19]. Growing evidence has suggested that systematic diseases such as diabetes and rheumatic inflammatory diseases are strongly associated with the development of OP [20,21]. In addition, glucocorticoids tend to induce iatrogenic osteoporosis [22]. Bone metabolism disorders and bone resorption occur more frequently than bone production in patients with osteoporotic edentulism. OP leads to elevated bone fragility and an increased risk of fractures. Moreover, the patients experience loss of bone density in the alveolar bone and are at an increased risk of tooth loss. Recently, with the development of oral implant technology, the demand for implants among people susceptible to tooth loss has increased rapidly [23]. However, implant treatment in such patients is often followed by problems of poor early implant stability, poor osseointegration, high incidence of peri-implant inflammation, and other complications, leading to failure of the implant treatment [24,25].

Bone is a metabolically active tissue that is constantly being repaired and remodeled by osteoblasts and osteoclasts. Any factor that decreases osteoblast activity or increases osteoclast activity can disrupt this bone metabolic balance, resulting in greater bone resorption by osteoclasts and reduced osteogenesis by osteoblasts, i.e., OP [26]. Increased levels of reactive oxygen species (ROS) affect many cellular processes and are associated with the development of aging and age-related diseases [27]. Substantial evidence suggests that ROS can increase bone resorption by directly promoting osteoclast differentiation and activity [28]. Additionally, they can induce apoptosis and reduce osteoblast activity, leading to reduced osteogenesis by osteoclasts [29]. Numerous studies have shown that oxidative stress, caused by overproduction of ROS, and inadequate antioxidant defense mechanisms [30] are key causative factors of OP [31].

As an antioxidant in the body, gastrodin maintains intracellular oxidative homeostasis in the physiological state, and its insufficient levels can lead to intracellular oxidative stress [32]. Postmenopausal women with estrogen deficiency show elevated ROS levels [33], and the plasma lipids exhibit hyperoxidative properties in vivo [34]. On the other hand, the activities of antioxidant-related superoxide dismutase (SOD) and catalase are reduced. These changes are strongly associated with decreased BMD in postmenopausal women [35,36]. In a mouse ovariectomy (OVX) model, OVX caused oxidative damage and reduced the capacity of antioxidant defense mechanisms, causing bone loss in the femurs of mice [37].

During the aging process, the body tissues and cells continuously produce free radicals, whereas the antioxidant component in the body decreases and the ability to scavenge free radicals diminishes. This leads to the accumulation of free radicals, which causes oxidative stress [38]. Oxidative stress has an important influence on the aging process and age-related diseases [39], while aging itself is a key factor for the decrease in bone mass and bone strength [40]. An association between oxidative stress and BMD decline in humans has been reported in numerous clinical studies [31]. In C57BL/6 mice, a progressive decline in bone strength and mass, with age, is temporally associated with increased apoptosis of bone cells such as osteoblasts. Notably, age-dependent decrease in bone mass and strength are associated with increased ROS levels [41].

Persistent hyperglycemia causes ROS overproduction by increasing mitochondrial oxygen consumption, disrupting mitochondrial function, or activating the ROS-producing enzyme nicotinamide adenine dinucleotide phosphate (NADPH) oxidase [42]. The levels of oxidative stress biomarkers are significantly elevated in the tissues and serum of type 2 diabetes mellitus (T2DM) patients [43]. ROS cause osteoblast oxidative damage by inhibiting the expression of antioxidant enzymes (nuclear factor erythroid 2-related factor [Nrf2] and hemoxygenase-1 [HO-1]) [44,45], increasing osteoblast apoptosis, by inhibiting the phosphoinositide-3-kinase (PI3K)/protein kinase B (AKT) signaling pathway [46], causingcellular ferroptosis by lipid peroxidation [44,47], and inducing osteoblast autophagy through the ROS-AKT-mammalian target of rapamycin axis [48]. Thus, it ultimately inhibits the osteogenic capacity of osteoblasts. Additionally, ROS can increase bone resorption by promoting osteoclast differentiation [49,50].

Thus, increased oxidative stress is a prominent feature in populations with a high prevalence of OP. Moreover, ROS causes bone loss by causing oxidative damage to the osteoblast and promoting osteoclast differentiation. As an antioxidant drug, gastrodin protects the osteogenic differentiation of osteoblasts [37,44], inhibits osteoclast differentiation [51], and alleviates osteoporotic symptoms in rats/mice [52].

## 3. Gastrodin

Gastrodin (Figure 1) is an extract found in dried tubers of the herb *Gastrodia elata* (*Tian ma*), which is a traditional Chinese medicine. Gastrodin is its main active ingredient, which is a phenolic compound [53], chemically known as 4-hydroxybenzyl alcohol-4-*O*-*β*-*D*-glucopyranoside, with molecular formula C13H18O7 and molecular weight 286 Da. In China, gastrodin has been used for centuries to treat headaches, dizziness, spasms, epileptic stroke, memory loss, and other diseases [54]. Modern medical research has shown that it not only has antioxidant, anti-inflammatory, and anti-apoptotic properties, but it also improves microcirculation, protects neurons, lowers blood pressure, and prevents osteonecrosis, among other functions. Currently, gastrodin is mostly used for the treatment of cardiovascular, cerebrovascular, and neuropsychiatric diseases [9,16,53,54,55,56] (Figure 2).

Gastrodin has many advantages in terms of pharmacokinetics and safety. First, it is rapidly absorbed into the intestinal tract. Gastrodin can be detected in plasma within 5 min of gavage administration (100 mg/kg) in rats [57]. In humans, after oral administration of gastrodin capsules (200 mg), the time to reach the maximum plasma concentration was 0.81 h [58]. Additionally, the bioavailability after oral administration of gastrodin in rats was approximately 80%; however, there were large differences between species [59,60]. Second, gastrodin is widely distributed after entering systemic circulation. After intravenous injection, gastrodin is distributed to the kidneys, lungs, liver, spleen, gastrointestinal tract, and brain in rats, and mainly to the kidneys, liver, and lungs in humans [61,62]. Third, acute and subacute toxicity experiments have proven that gastrodin has a high safety factor and high potential for continuous administration of larger doses [63,64,65]. Clinical adverse reactions/events caused by gastrodin have occasionally been reported, with the main symptoms including rash, pruritus, dizziness, dry mouth, nausea, palpitations, vomiting, and headache [66].

## 4. Gastrodin for Osteoporosis Treatment

An increasing number of studies have demonstrated the potential of gastrodin in the treatment of OP. Through its antioxidant, anti-apoptotic, and anti-inflammatory properties, gastrodin promotes the viability and osteogenic differentiation of osteoprogenitor cells, preosteoblasts, and periodontal stem cells, and inhibits osteoclast differentiation, thereby improving bone formation and reducing bone loss. The relevant studies are summarized in Table 1.

### 4.1. Protection of Osteogenesis

#### 4.1.1. Antioxidant Effect

Mitochondria are the powerhouses in the cellular microenvironment and provide an impetus for cell survival and function. Bone formation and bone remodeling require significant energy consumption, and the energy produced by the mitochondria is essential for maintaining the growth, differentiation, and biological functions of the osteocytes [71]. Any interference with the oxidative phosphorylation pathway in the mitochondria impairs osteogenic gene expression and extracellular matrix (ECM) synthesis in C3H10T1/2 mesenchymal progenitor cells [72]. However, oxidative stress can also lead to cellular dysfunction. When several ROS are generated in the cell, they attack the mitochondrial membrane and mitochondrial DNA, which enhances the autophagy of mitochondria and changes the permeability of the outer mitochondrial membrane. These behaviors disrupt the mitochondrial structure and function, resulting in reduced cellular adenosine triphosphate production and increased ROS production, eventually triggering mitochondrial and cellular dysfunction [73].

The mitochondrial ROS balance is achieved by mitochondrial antioxidants, including Nrf2 [74]. Gastrodin effectively scavenges oxygen radicals to exert antioxidant activity, downregulates lipid peroxidation levels, inhibits uncoupled oxidative phosphorylation, and increases the expression of genes encoding antioxidant proteins such as Nrf2 and HO-1 [9,75]. By upregulating the expression of the Nrf2/KEAPl antioxidant pathway (NRF2, HO-1, and NADPH quinone oxidoreductase-1), gastrodin reduces the dexamethasone-induced oxidative stress levels in MC3T3-E1 cells and mitochondria, increases osteoblast viability, promotes the expression of osteogenesis-related markers such as Runx2, osterix, bone morphogenetic protein (BMP) 2, and osteocalcin (OCN), and improves the alkaline phosphatase (ALP) activity and osteogenic mineralization capacity. On the other hand, the antioxidant protective effect of gastrodin is diminished by knocking out Nrf2 [44,67].

Both bone marrow-derived mesenchymal stem cells (BMSCs) and osteoblasts are involved in bone formation, with the former mainly differentiating into osteoblasts or adipocytes [76]. In vitro studies suggest that oxidative damage may partly contribute to OP by inhibiting the osteogenic differentiation of BMSCs [77]. In elderly patients with OP, the decrease in BMD is accompanied by a decrease in osteoblasts and increase in adipocytes, suggesting that the balance between osteogenic differentiation and lipogenic differentiation of BMSCs is one of the important factors affecting bone quality [78,79]. Gastrodin inhibits H_2_O_2_-mediated overproduction of ROS in human bone marrow stromal stem cells (hBMMSCs), significantly promotes the proliferation of hBMMSCs, upregulates the expression of the osteogenic genes ALP, BGLAP, and COL1A1, protects cellular ALP activity and calcification mineralization, and reduces the expression of the lipogenic genes CFD and LPL. Eventually, gastrodin promotes osteogenic differentiation and inhibits lipogenic differentiation of hBMMSCs under oxidative stress [37].

Sirtuin 3 (SIRT3) is a protein deacetylase member of the sirtuin family that is located mainly in the mitochondria. SIRT3 is involved in energy metabolic processes, including the respiratory chain, tricarboxylic acid cycle, fatty acid β-oxidation, and ketogenesis. Thus, SIRT3 controls the flow of the mitochondrial oxidative pathway and the rate of ROS production [80]. It can also affect malondialdehyde (MDA) levels [69], which is an important marker of oxidative stress. It has been reported that SIRT3-deficient mice present OP [81]. Human periodontal ligament stem cells (hPDLSCs) can differentiate into alveolar bone and periodontal ligament-like tissues and in multiple directions [82]. In a lipopolysaccharide (LPS)-induced oxidative damage model of hPDLSCs, gastrodin inhibited oxidative stress in hPDLSCs by upregulating SIRT3 gene expression and decreasing the levels of MDA and lactate dehydrogenase, which are markers of oxidative stress. Gastrodin significantly promoted hPDLSC’s proliferative viability and ALP activity, mineralized nodules, and increased the expression of the osteogenic differentiation-related proteins ALP, Runx2, OCN, and osteopontin [69].

In vivo, gastrodin reduced oxidative stress, promoted osteogenic differentiation and mineralization processes, and enhanced bone microstructure and biomechanical strength in glucocorticoid-treated osteoporotic rats [67]. In OVX mice and T2DM rat OP models, gastrodin significantly reduced serum MDA activity, increased glutathione and SOD activity, enhanced antioxidant status, and alleviated bone loss [37,68]. Gastrodin reduced serum MDA levels, increased SOD activity, reduced ROS accumulation, and alleviated femoral and alveolar bone damage in rats with fluorosis [9].

#### 4.1.2. Anti-Apoptotic Effect

Apoptosis induction is closely associated with the release of apoptotic factors such as Bax, cytochrome C, pro-caspases, and apoptosis-inducing factor (AIF) [83]. When the mitochondria are subjected to stress disorders, Bax aggregates and oligomerizes at different sites in the mitochondria and regulates cytochrome C translocation release [84]. Cytochrome C release initiates caspase-protease-dependent apoptosis. Additionally, AIF can induce chromosomal DNA-independent cell division and enhance apoptosis. However, gastrodin can block this cascade response, reduce the protein expression of Bax, cytochrome C, caspase-3 and AIF, and increase the production of the anti-apoptotic factor Bcl-2 to inhibit apoptosis in osteoblasts [44,67]. In a chondrocyte-mimicking in vitro osteoarthritis model, gastrodin attenuated interleukin (IL)-1β-induced chondrocyte apoptosis by inhibiting the nuclear factor kappa B (NF-kB) signaling [84]. LPS stimulation significantly decreased the expression of Bcl-2 and increased the expression of Bax, caspase-3, and caspase-9. However, gastrodin pretreatment inhibited the LPS-induced apoptosis of the hPDLSCs [79].

In vivo, gastrodin reduced Bax, caspase-3, and caspase-9 protein expression levels and increased Bcl-2 expression in rats with fluorosis [9]. Gastrodin reduced the incidence of osteonecrosis by exerting an anti-apoptotic effect in rats with osteonecrosis [52]. It also improved the peri-implant cancellous bone quality through an anti-apoptotic effect in T2DM rats [68]. Furthermore, gastrodin improved the balance of expression between apoptotic and anti-apoptotic factors in osteoarthritic rats, increased the deposition of proteoglycans in the ECM, and reduced damage to the subchondral bone plate [70].

#### 4.1.3. Anti-Inflammatory Effect

NF-kB signaling, one of the most important intracellular signaling pathways, is involved in the regulation of inflammatory and pro-inflammatory stress-related responses [85]. When stimulated by inflammatory factors, NF-kB is activated to undergo nuclear translocation and trigger the transcription of inflammation-related genes. Gastrodin-attenuated NF-kB nuclear translocation in chondrocytes reduces the ratio of p-IkB-α/IkB-α, decreases the expression of the inflammatory factors such as tumor necrosis factor (TNF)-α and IL-6, reduces the degradation of the ECM and matrix metalloproteinase 3, and maintains intracellular homeostasis in the chondrocytes. Gastrodin improved cartilage degeneration in an osteoarthritis rat model in vivo [70]. In LPS-induced injury of hPDLSCs, gastrodin significantly reduced the expression of TNF-α and IL-6 and alleviated inflammatory injury [83]. Moreover, gastrodin inhibited the expression of TNF-α and IL-6 in hBMMSCs under oxidative stress [37], and factors such as receptor activator of NF-kB ligand (RANKL), TNF-α, and IL-6 are highly involved in estrogen-deficient OP cases [86].

### 4.2. Inhibition of Bone Resorption

#### 4.2.1. Gastrodin Inhibits Osteoclast Differentiation under Oxidative Stress through Antiox-Idant Effect

Osteoclasts are derived from a monocyte/macrophage cell line (RAW264.7 cells). They are mainly involved in bone resorption and can secrete hydrochloric acid and lysozyme extracellularly to destroy and dissolve the surrounding bone tissue. The normal function of osteoblasts and osteoclasts is to maintain the homeostasis of bone metabolism [87]. Substantial evidence suggests that ROS can increase bone resorption by directly promoting osteoclast differentiation and activity [28]. However, gastrodin reduces the level of ROS in RAW264.7 cells under oxidative stress, inhibits increased osteoclast-specific gene expression (NFATc1, TRAP, CTR, and CTSK) induced by H_2_O_2_, and reduces the number of osteoclasts. Thus, gastrodin may exert potential anti-osteoporotic effects by inhibiting osteoclast differentiation [37].

#### 4.2.2. Gastrodin Inhibits Osteoclast Differentiation in Normal Environment through Antioxidant Effect

Nuclear factor of activated T cells cl (NFATc1) plays a key role in osteoclast differentiation. RANKL activates NFATc1 expression through a series of cascade signals (recruitment of TNF receptor-associated factor 6, mitogen-activated protein kinase, AKT, and NF-kB pathway), which leads to the differentiation and maturation of osteoblasts [88]. Exogenous NFATc1 can still induce osteoclast differentiation in the absence of RANKL, whereas NFATc1-deficient embryonic stem cells cannot differentiate into osteoclasts in the presence of RANKL [89]. However, gastrodin effectively delays the differentiation of the bone marrow-derived macrophages (BMMs) into osteoclasts by downregulating the transcriptional and translational expression of NFATc1. The expression of osteoclast-specific genes, such as TRAP, Cts K, and DC-STAMP, is significantly reduced by gastrodin [51].

Osteoclast differentiation is a multistep process that involves cell proliferation, commitment, fusion, and activation, and the migration of pro-osteoclasts is necessary during the fusion process. In wound-healing experiments, gastrodin significantly inhibited the migration of pro-osteoclasts. Moreover, the bone resorption by osteoclasts in bone fragments was inhibited by gastrodin intervention [51].

In a healthy organism, ROS at normal levels participate in the regulation of normal operation of various biological functions. The differentiation of osteoclasts requires the activation of RANKL, and this process needs the involvement of moderate ROS [28]. Compared with other bone cells, osteoclasts need more ROS [90]. Therefore, we speculate that in normal environments, such as this study, gastrodin still inhibits osteoclast differentiation by reducing ROS levels. This low level of ROS is insufficient to maintain the activation of RANKL, which is required for osteoclast differentiation.

## 5. Gastrodin and Actin Filament in Anti-Osteoporosis

### 5.1. Gastrodin Can Protect Actin Filament under Oxidative Stress

The actin cytoskeleton must be finely tuned, which is energetically costly, both in space and time, to fulfill key cellular functions such as cell division, cell shape changes, phagocytosis, and cell migration [91,92]. For example, during osseointegration, the migration, adhesion, and morphology of osteocytes on the surface of the material depend heavily on the assembly/de-assembly of actin filaments. Furthermore, the actin cytoskeleton is a key regulator of apoptosis and ageing. Thus, more attention should be paid to the actin cytoskeleton in bone and osteocyte related studies.

An appropriate or normal production of ROS has been demonstrated to function as an important signaling component, which plays a role in remodeling the actin cytoskeleton. For example, 1.5 mM H_2_O_2_ treatment induced the recruitment of the Arp2/3 complex at the leading edge of PtK1 cells, accelerating the nucleation of G-actin [93]. However, direct oxidation of actin by excess ROS has been shown to affect polymerization. Treatment of G-actin with H_2_O_2_ (5 mM) decreased its polymerization ability. And actin filaments polymerized from H_2_O_2_-oxidized monomers were more fragmented and fragile, compared to unoxidized filaments [94,95]. Similarly, in the report by Lin et al. [96], MC3T3-E1 cells undergoing oxidative stress had poor adhesion morphology on the tantalum surface and poor actin filament structure. In other studies, gastrodin reduced intracellular ROS content and improved the migration and adhesion morphology of human umbilical vein endothelial cells (HUVECs) and astrocytes by antioxidants [17,97,98,99], while gastrodin protected the structure and function of mitochondria under oxidative stress to maintain ATP production [44,100,101,102]. In the following, we speculate on the potential mechanisms of gastrodin to protect cellular actin filaments and the potential of gastrodin with Mg^2+^ to improve cellular function.

### 5.2. Effects of Oxidative Stress on Actin Filament Polymerization-Depolymerization

Actin filaments are assembled from actin monomers (G-actin). The number of free G-actins is sufficient for actin filament assembly. However, only ATP-bound G-actin can participate in actin filament assembly. The ATP-bound G-actin assembles at the actin filament positive extreme, with the involvement of Mg^2+^, at which point the combined ATP hydrolyze to ADP and the actin filament tends to polymerize and lengthen. In an environment with an appropriate concentration of Ca^2+^, the G-actin tends to disassemble at the negative extreme, and the actin filament tends to depolymerize and shorten. Actin filaments maintain normal structure and function in a continuous polymerization-depolymerization dynamism. Nucleation is the rate-controlled step in the assembly of G-actin in vitro. Stimulated by foreign signals, the nucleating protein Arp2/3 complex binds to the cell membrane or other structures, providing a binding site for G-actin, thus greatly accelerating the nucleation process. The Arp2/3 complex provides the opportunity for G-actin to start assembling, much like the first locomotive responsible for traction. The Arp2/3 complex can also bind to existing actin filaments, initiating the assembly of new actin filaments to form a tree-like grid [103].

Notably, the consequences of H_2_O_2_ treatment to actin filaments strongly depend on the nature and concentration of the divalent cations, as well as the nucleotide that is bound to G-actin [91]. H_2_O_2_ (5 mM) abolishes the polymerization ability of ADP-bound G-actin on actin filaments, which accelerates the depolymerization of actin filaments, but has no such effect on ATP-bound G-actin [95]. Treatment of Ca^2+^-bound G-actin with H_2_O_2_ decreases its polymerization extent and accelerates actin filament depolymerization [94,95]. Compared to Ca^2+^-bound G-actin, Mg^2+^-bound G-actin is resistant to similar concentrations of H_2_O_2_ [104,105].

### 5.3. The Potential Mechanism of Gastrodin Protecting Actin Filament and the Combination of Gastrodin and Mg^2+^

Therefore, we speculate that, on the one hand, gastrodin reduces intracellular ROS content through the antioxidant effect, alleviating the oxidative damage of G-actin, the polymerization function is protected, and the depolymerization rate of the negative extreme on actin filament is slowed down. At the same time, ROS content decreased close to normal levels, at which it could induce the recruitment of the Arp2/3 complex. On the other hand, gastrodin protects the mitochondrial function to produce ATP, increases the amount of ATP-bound G-actin, and improves the polymerization rate of the positive extreme on actin filaments. In addition, a certain concentration of Mg^2+^ is one of the necessities for G-actin polymerization, which also enhances the resistance of G-actin to H_2_O_2_. Thus, the application of gastrodin combined with Mg^2+^-rich preparations may modulate the polymerization-depolymerization dynamic balance of actin filaments in a peroxidizing environment (Figure 3). This approach may have great potential to improve osteocyte function during bone regeneration and osseointegration, especially oxidative stress, concomitantly. Of course, even in a healthy organism, the implantation of bone regeneration or osseointegration materials is often accompanied locally by inflammation, oxidative stress, etc.

In addition, BMSC cells in the gastrodin group showed a clustered morphology with more actin filaments linking adjacent cells in the absence of oxidative stress [51]. Given that actin filament assembly/de-assembly will eventually reach dynamic equilibrium in a normal physiological state, specifically, the polymerization rate of the positive extreme is equal to the depolymerization rate of the negative extreme. We speculated that the possible mechanism by which gastrodin directly increases the actin filament numbers of BMSC cells is that gastrodin accelerates the nucleation process of G-actin, and initiates more actin filament polymerization by regulating the upstream signal of the Arp2/3 complex. However, in a study of anti-depressant in rats, gastrodin downregulated the expression of the neuronal cytoskeleton remodeling-related negative regulators Slit1 and Ras homologous member A (RhoA), but upregulated the positive regulators dihydropyrimidinase-related protein 2 (CRMP2) and profilin 1 (PFN1). In addition, slit1 expression in the Human oligodendroglioma cell line (Hs683) was directly down-regulated by gastrodin [106].

Based on the remarkable significance of actin cytoskeletons for cell life activity, the beneficial effects of gastrodin on the actin cytoskeleton should be considered. More research is needed to discover and prove the mechanisms involved in this process.

## 6. Gastrodin and RAS in Anti-Osteoporosis

### 6.1. Effects of RAS and RAS Inhibtors on OP

The classical renin–angiotensin system (RAS) is a key regulator of blood pressure, water, and electrolytes, and maintains their stability in the internal environment [107]. The RAS consists of angiotensin II (AngII), angiotensin-converting enzyme (ACE), angiotensin type 1, 2, 3 and 4 receptors (AT1R, AT2R, AT3R and AT4R), ACE structural homolog ACE2, angiotensin-(1–7) peptide, and Mas receptor. Renin cleaves angiotensinogen to produce angiotensin I, which is further cleaved by ACE to generate Ang II. Then Ang II binds to specific receptors (AT1R, AT2R), triggering a broad range of biological actions [108]. RAS is widely studied for the treatment of hypertension, heart failure, and kidney disease. However, the role of various RAS components in bone remodeling and metabolism have received increasing attention, lately [109]. The local RAS was found to participate in age-related OP of aging mice [110], and the development of glucocorticoid-induced OP of rabbits [111]. In addition, RAS accelerated OP in ovariectomized rats [112]. In vitro studies showed that Angiotensin II suppresses osteoblastic differentiation and mineralized nodule formation via the AT1 receptor in osteoblasts [113]. Ang II was also shown to accelerate osteoclastic functions. Meanwhile, Ang II enhanced Ti-particle-induced suppression of osteogenic differentiation in mouse BMSCs. In addition, local bone RAS promoted osteolysis by increasing bone resorption and decreasing bone formation by modulating the RANKL/RANK and Wnt/β-catenin pathways [114,115].

More recently, RAS inhibitors have been confirmed to have beneficial effects on bone tissue [116]. ACE inhibitors (ACEIs), imidapril or perindopril, improved bone structure in rats and mice by decreasing bone resorption and increasing bone formation [114,117]. Renin inhibitor, aliskiren, exhibited the beneficial effects on trabecular bone of ovariectomy-induced osteoporotic mice [112].

Ang II mainly relies on AT1R to mediate its biological responses [109]. A study found that Ang II decreased the expression of Runx2, Msx2, and osteocalcin in ROS17/2.8 cells through the AT1R. The AT1R blocker (ARB1), losartan, completely inhibited this effect [113]. Another study found that both ARB1 (olmesartan) and ARB2 (PD123319) increased BMD in C57BL/65 mice, following OVX, by inhibiting osteoclastic activity. Notably, ARB1 showed more capacity than ARB2 in relieving Ang II-induced osteoporosis [118]. Similarly, the administration of olmesartan improved Ang II-induced decreased bone mass in another study [115]. Interestingly, Knockdown of AT2R with siRNA markedly inhibited osteoclast formation, stimulated by Ang II. However, knockdown of AT1R resulted in a further increase in osteoclast formation by Ang II, suggesting that AT1R may exert an inhibitory effect on AT2R. The functions of AT1R and AT2R are in many cases counter-regulatory to each other [119]. Of course, the authors also agree that such results cannot rule out the influence of the known sex difference of AT2R expression in the Tsukuba hypertensive mouse.

### 6.2. Effects of Gastrodin on RAS

The beneficial effects of gastrodin on neuroinflammation, hypertension and heart disease by inhibiting RAS deserve our attention. The relevant studies may give us new enlightenment on the treatment of osteoporosis with gastrodin. Ang II via AT1R stimulation can activate NADH oxidase to inhibit the expression of SIRT3. However, in activated microglia in the corpus callosum of hypoxic-ischemic brain damage (HIBD) rats, and in LPS stimulated BV-2 microglia, gastrodin blocked RAS by inhibiting the activation of ACE and AT1R, eventually decreasing the expression of nicotinamide adenine dinucleotide phosphate oxidase-2 (NOX-2) and TNF-α and increasing expression of AT2R and SIRT3 [120]. Similarly, Wu suggested that gastrodin acts via the downregulation of RAS, which regulates the Notch-1 signaling and inflammation in LPS-induced microglia [121]. A bioinformatics analysis revealed that Ang II-mediated mouse myocardial hypertrophy, reversed by gastrodin, were associated with regulating RAS protein signal transduction [122]. Furthermore, another study showed that gastrodin inhibited the renin-angiotensin-aldosterone system (RAAS) to decrease the systolic blood pressure (SBP) in spontaneously hypertensive rats (SHRs) [123].

### 6.3. Gastrodin May Inhibit OP via RAS, and ARB2 May Be an Adjunct

Therefore, it is meaningful to explore the mechanism of gastrodin influences on OP, through RAS. Firstly, as mentioned in previous studies, gastrodin protects bone formation through anti-oxidation, by upregulating the expression of SIRT3, [69] and anti-inflammation, via the downregulation of RANKL [70]. These effects have been confirmed. However, the authors have not given answers to the upstream mechanism of gastrodin regulation of SIRT3 and RANKL. Based on the effects of RAS and RAS inhibitors on bone metabolism mentioned in this paper, and the regulation of gastrodin on RAS and RAAS, we suggest that RAS may be part of the upstream action target of gastrodin to inhibit OP through anti-oxidation and anti-inflammation. Secondly, gastrodin inhibits the activation of ACE and AT1R but increases the expression of AT2R [120]. If Knockdown of AT2R with siRNA markedly inhibited osteoclast formation, as Asaba et al. [119] pointed out, we may be able to combine gastrodin with an appropriate amount of ARB2. Thirdly, other downstream signal changes caused by gastrodin through RAS may play a role in bone metabolism, which is worth exploring and studying (Figure 4).

## 7. Gastrodin and Ferroptosis in Anti-Osteoporosis

### 7.1. Gastrodin May Inhibit OP by Anti-Ferroptosis

Ferroptosis is an iron- and ROS- dependent form of regulated cell death, characterized by cystine starvation, glutathione consumption, iron overload, and related lipid peroxidation [47,124]. Recently, it was found that bone loss can be alleviated by inhibiting ferroptosis via anti-oxidation. Activation of the Nrf2/HO-1 signaling pathway reduced ferroptosis to suppress diabetic OP [47,125]. Bone loss in iron overload mice induced by iron dextran was ameliorated via the activation of the Akt/GSK3β/Nrf2 pathway [126]. Upregulation of FoxO1 and Nrf2 levels inhibited excess-iron-induced bone loss in mice and MC3T3-E1 Cells [127]. The cytotoxic effect of iron overload induced by ferric ammonium was inhibited via activation of Nrf2/ARE signaling in MC3T3-E1 cells [128]. Recent research showed that gastrodin improved neurological dysfunction in cellular or animal ferroptosis models through the NRF2/HO-1 pathway, including the glutamate-induced hippocampal neuron (HT-22) cell ferroptosis model [129], erastin-induced HT-22 cell ferroptosis model, cognitive dysfunction in vascular dementia rats [130] and the complete Freund’s adjuvant-induced chronic inflammatory pain model in mice [131]. Therefore, we suggest that gastrodin may inhibit ferroptosis in osteoblasts through the antioxidant effect, thereby protecting osteoblasts viability and alleviating OP.

However, it should be noted that although gastrodin can inhibit the differentiation of osteoclasts, gastrodin may also protect osteoclast viability via inhibition of ferroptosis, especially when osteoclasts are already under oxidative stress. Moreover, it has been proved that gastrodin protects the viability of pre-osteoclasts by reducing intracellular oxidative damage [37]. In the two related studies, gastrodin does not inhibit the proliferation of BMMs [51] but, protects the proliferation of RAW264.7 cells under oxidative stress [37]. This shows us that counteracting this relatively positive effect of gastrodin on osteoclasts (which is relatively negative for anti-osteoporosis) is conducive to amplifying the anti-osteoporosis effect of gastrodin.

### 7.2. Artemisinin May Selectively Inhibit Osteoclast

Artemisinin (ARS), a sesquiterpene lactone compound, is a traditional Chinese medicine ingredient isolated from a herb [90]. In addition to antimalarial activity, ARS compounds, such as artesunate (ART), artemether (ARM) and dihydroartemisinin (DHA), exhibit the potential to treat bone loss in accumulating studies [132,133,134,135,136]. The release of large amounts of free radicals by activated ARS compounds, which cause oxidative damage, relies on iron-activated artemisinin peroxide groups [137]. Additionally, osteoclast intracellular iron content is higher than other normal cells [138]. Therefore, Zhang et al. [90] creatively put forward that the high level of intracellular iron makes osteoclasts become the potent targets of ARS compounds, which can selectively inhibit the differentiation and viability of osteoclasts by inducing intracellular oxidative damage and ferroptosis, but not osteoblasts. In fact, in the current studies, ARS and its derivatives do not affect, or have a very limited role in, the differentiation of osteoblasts [90] and BMSCs [134], but can inhibit the differentiation and formation of osteoclasts [139,140,141].

### 7.3. The Potential of Gastrodin and ARS in Combination against OP

Based on the possible protective effects of gastrodin on the viability of osteoclasts under oxidative stress [37], the selective inhibitory effect of ARS on osteoclasts [90] and the known osteoprotective effect of gastrodin, shown in Table 1, we propose that the combination of gastrodin and ARS may have greater therapeutic potential for OP. This drug combination is characterized by protecting the proliferation and differentiation of pre-osteoblasts [37,44], inhibiting the differentiation and maturation of pre-osteoclasts [37,51], and further selectively inhibiting the viability of osteoclasts which can counteract the relatively positive effect of gastrodin on osteoclasts [90,139,140,141] (Figure 5). First of all, it is necessary to verify the protective effects of gastrodin on the viability of osteoclasts, through reduction of oxidative damage under oxidative stress (the protection on the viability of pre-osteoclasts has been proved [37]). Through this step, we will determine that the combination with ARS, which selectively inhibits osteoclasts, is ingenious and meaningful. Secondly, verifying the inhibition of gastrodin on ferroptosis of osteoblasts and osteoclasts can further indicate the value of this drug combination. Finally, it is important to explore the appropriate dosage combination of the two drugs.

## 8. Gastrodin and Tissue Engineering Bone Regeneration

In tissue engineering, gastrodin is loaded into the scaffold, where it exerts antioxidant properties, promotes the process of bone repair and regeneration by increasing the hydrophilicity of the material surface, conducts immunomodulation, and promotes vascular renewal. Additionally, gastrodin has a relatively simple chemical structure and polyhydroxy functional groups, which are favorable for chemical polymerization [13,14,15]. Its pharmacokinetic characteristics and wide biodistribution are favorable for continuous drug delivery. It can increase the hydrophilicity of the material surface, promote degradation without decreasing the compressive strength and elastic modulus of the material, increase the cross-link density of the composite scaffold, and improve functional reconfiguration [13,142]. The relevant studies are summarized in Table 2.

### 8.1. Gastrodin Maintains Its Inherent Antioxidant Property

In previous studies, gastrodin improved bone metabolic homeostasis through its antioxidant properties either by gavage in animal models or by addition to a medium to stimulate cells. It is important to assess whether this antioxidant activity is maintained when gastrodin is loaded into scaffold films applied in tissue engineering. Zheng et al. [17] fabricated gastrodin/polyurethane (Gastrodin/PU) films using a solvent casting/salt leaching process. Specifically, the Gastrodin/PU material was dissolved in 1,4-dioxane in a 1:2 ratio with NaCl salt to form a paste and then cast into Teflon molds. After drying, the salt component of the scaffold was removed by immersion in deionized water to form a porous film. In follow-up experiments, for H_2_O_2_-treated human umbilical vein endothelial cells (HUVECs), the Gastrodin/PU scaffold showed significant free radical scavenging activity when compared to the PU scaffold, effectively upregulating the expression of HO-1 and Nrf2 in HUVECs and protecting them from oxidative damage. The free radical scavenging ability positively correlated with the release rate of gastrodin. Nevertheless, the maintenance of this antioxidant activity of gastrodin needs to be verified in bone tissue experiments.

### 8.2. Gastrodin Improves the Hydrophilicity of the Material Surface

The adhesion of osteoblasts to biomaterials is not simply physical adhesion but a complex process that initiates and regulates cell survival, migration, recruitment, and osteogenic differentiation [145]. Osteocyte adhesion is often enhanced in bone tissue engineering using various methods to promote bone repair and regeneration [146]. Most studies have shown that hydrophilic surfaces are more conducive to cell adhesion. The adhesion ability of MC3T3-E1 cells to the surface of a polycaprolactone scaffolds was significantly enhanced when the hydrophilicity of the surface was increased [147]. The adhesion ability of SAOS-2 osteoblasts was inhibited on the hydrophobic side of the biopolymer but not significantly on the hydrophilic side [148]. Therefore, osteoblast adhesion can be controlled by modulating the hydrophilicity of the material surface [149].

The Gastrodin-PU/nano-hydroxyapatite (n-HA) scaffold obtained by the in-situ foaming method [142] has desirable hydrophilic properties without degrading the physical properties of the material. The hydrophilicity of the scaffold surface increased with an increase in the gastrodin content [13]. In vitro, in addition to the inherent antioxidant properties of gastrodin, the gastrodin-PU/n-HA scaffold promoted the adhesion and migration ability of rat bone marrow mesenchymal stem cells (rBMSCs) on the scaffold by increasing the hydrophilicity [13]. Similarly, the Gastrodin/PU scaffold had higher hydrophilicity than the PU scaffold, HUVECs, Schwann cells, and PC12 cells, which all showed improved adhesion to the Gastrodin/PU scaffold surface [17,143,144]. In vivo, implantation of scaffolds loaded with gastrodin in the bone or under the skin promoted the repair and regeneration of bone, blood vessels, and nerves [13,17,119].

Additionally, numerous studies have shown that increasing the surface hydrophilicity of biomaterials ultimately promotes the initial formation and attachment of proteins, activates relevant signaling pathways, induces high expression of ALP/OCN, and ultimately promotes the differentiation of MC3T3 cells [150,151,152]. Li et al. [13] mentioned that the high surface hydrophilicity of materials also facilitates macrophage polarization to the M2 phenotype, which exerts anti-inflammatory effects. On the other hand, Yang et al. [153] suggested that the M2 polarization of the macrophages might be attributed to the combined effects of roughness and hydrophilicity.

### 8.3. Anti-Inflammatory Effect of Gastrodin

The use of different materials or modification of the material surface can enhance bone formation in vivo by directly stimulating the osteogenic functions of osteoblasts and preosteoblasts in vitro [154]. However, bone healing is a complex process of interacting biological factors involving inflammatory immune regulation, angiogenesis, and osteogenic differentiation [155,156]. After implantation, the biomaterial interacts with the immune cells and triggers an inflammatory response. An unrestricted inflammatory response can disrupt the bone metabolic balance, leading to a delay in bone regeneration and affecting the success of long-term implantation of the material [157,158]. In the case of in vivo bone grafting, macrophages may be among the first immune cells reaching the implant [159], and they play a central role in coordinating the immune inflammatory responses [160,161]. Macrophages can polarize towards a pro-inflammatory phenotype (M1 phenotype) or a pro-healing phenotype (M2 phenotype). Macrophages of the M2 phenotype typically express high levels of IL-10, BMP-2, and vascular endothelial growth factor (VEGF), which suppress inflammation, recruit bone progenitor cells, and activate angiogenesis and bone regeneration [162,163]. In the study by Li et al., compared to the PU/n-HA scaffold, the Gastrodin-PU/n-HA scaffold induced macrophage polarization to the M2 phenotype with higher expression of pro-regenerative cytokines (CD206 and Arg-1) and lower expression of pro-inflammatory cytokines (iNOS). Furthermore, the expression levels of osteogenesis-related factors (BMP-2 and ALP) in rBMSCs and angiogenesis-related factors (VEGF and basic fibroblast growth factor) in HUVECs were significantly upregulated in the gastrodin-PU/n-HA/macrophage-conditioned medium compared to those in the PU/n-HA/macrophage-conditioned medium. This indicates that gastrodin-PU/n-HA-induced macrophage M2 polarization drives osteogenic differentiation of rBMSCs and vascular regeneration. In vivo, a 2% gastrodin-PU/n-HA scaffold implanted in a rat femoral condylar defect model accelerated osteogenesis and angiogenesis [13]. It was demonstrated that gastrodin loaded into the scaffold promoted bone repair and vascular regeneration by its involvement in inflammatory immune regulation and pro-macrophage M2 polarization.

### 8.4. Gastrodin Promotes Vascular Regeneration

Vascularization plays a crucial role in bone regeneration. The vasculature provides nutrients, growth factors, minerals, and oxygen for tissue regeneration to sustain cell survival and interaction, diverts waste products from the healing zone, and releases paracrine signals that regulate cell growth, differentiation, and regeneration [164]. For instance, VEGF-A, a member of the VEGF family, promotes the proliferation and migration of the endothelial cells, regulates the secretion of osteogenic growth factors, and stimulates osteogenesis through paracrine signaling. VEGF is also known to promote the repair and regeneration of bone defects [165,166,167].

Numerous studies have demonstrated the promoting and protective effects of gastrodin on vascular neogenesis. Gastrodin increased the level of VEGF in rat serum and promoted vascular regeneration in damaged tissues [168,169]. Similarly, in a zebrafish model, gastrodin was found to promote angiogenesis [170]. In vitro, gastrodin either attenuated tert-butyl hydroperoxide-induced apoptosis and dysfunction in HUVECs by activating the Nrf2/HO-1 pathway [97] or, activated the PI3K/AKT signaling pathway by increasing miR-21 expression in the HUVECs [98]. Both processes can improve the proliferation, migration, and cell tube formation of HUVECs and induce neointima formation. In tissue engineering, gastrodin is loaded onto various types of PU scaffolds, which increases the material surface hydrophilicity and facilitates immunomodulation and anti-inflammation, all of which promote high expression of angiogenic-related factors that effectively enhance the vascular tissue regeneration after implantation [13,17,143]. The effective promotion of angiogenesis by gastrodin induces bone repair and regeneration [13].

## 9. Gastrodin and Implant Osseointegration

### 9.1. Studies on the Promotion of Osseointegration by Gastrodin

Osteoblast adhesion is the initial step in early osseointegration of implants, influencing behaviors such as cell migration, proliferation, and differentiation. Moreover, it is critical in determining early osseointegration between the host bone tissue and implant material. Enhancing osteoblast adhesion is one of the primary goals for optimizing the surface properties of bone biomaterials [149]. Oxidative stress inhibits osteoblast adhesion to the material surface. The higher the oxygen content of the tantalum surface, the more ROS production is induced within MC3T3-E1 osteoblasts on its surface. Compared to cells with less ROS and MC3T3-E1, osteoblasts with excessive ROS had reduced number of adhesions on the tantalum surface, poor adhesion morphology, and reduced osteogenic differentiation and mineralization capacity. In vivo, the highly oxygenated tantalum material is surrounded by a reduced amount of new bone and poor osseointegration [96]. In a study by Zhang et al. [68], gastrodin improved the bone tissue microstructure around rat femoral implants, promoted osseointegration by regulating the oxidative stress state and balance between the expression of the apoptotic and anti-apoptotic factors in bone tissue in vivo, and reduced blood glucose levels in T2DM rats. This indicates that gastrodin may promote implant osseointegration via its antioxidant action.

In another study, compared to untreated BMSCs, gastrodin promoted the adhesion of BMSCs to the titanium surface and increased ALP activity [51]. The relevant studies are summarized in Table 2.

Based on the aforementioned two studies on gastrodin and osseointegration that could be found, gastrodin has the potential to promote osseointegration of implants. However, limited by the number and content of studies, the mechanism of action should be explored and clarified further in a comprehensive and detailed manner.

### 9.2. Prospect on the Application of Gastrodin for Osseointegration

Traditional systemic-assisted drug delivery methods have many disadvantages, such as high-dosage, adverse reactions, and poor local effects, whereas artificial self-loaded dental implants with locally released drug delivery system can improve the implant success rate. Drug loading on the surface of oral implants can be achieved by surface modification and fabrication of drug-loaded biological coatings [155]. The drug is loaded onto the coating by physical adsorption and chemical binding and released from the implant by diffusion, penetration, or degradation [171].

Gastrodin has a relatively simple chemical structure, can increase the hydrophilicity of the material surface, and can promote bone tissue repair and regeneration in various ways. These characteristics of gastrodin have been well applied and exploited in tissue engineering research. Therefore, we speculate that the application of gastrodin for oral implant surface modification or drug-loaded bio-coating may greatly promote implant osseointegration and improve early stability and long-term implant success rate (Figure 6). However, there is a paucity of research on the application of gastrodin in oral implant surface loading.

## 10. Conclusions

Gastrodin is found to have anti-osteoporosis effects. Its mechanisms of action include its antioxidant effect, anti-inflammatory effect, anti-osteoblast apoptosis, inhibition of osteoclast differentiation. Furthermore, given the many unique advantages of gastrodin in promoting bone healing in tissue engineering, such as inducing high hydrophilicity in the material surface, anti-inflammatory effect, and pro-vascular regeneration, we suggest that the use of gastrodin in the surface loading of oral implants may have great potential and effectively promote implant osseointegration. This assumption may provide inspiration for improving the success rate of patients’ implants in clinical practice. However, there is a paucity of research on gastrodin as an auxiliary drug for oral implants osseointegration. Particularly, the research on gastrodin involved in local sustained-release systems on the surface of oral implants is completely blank.

In addition, based on the existing research, we also hypothesized the potential mechanisms of gastrodin affecting actin filament polymerization, RAS and ferroptosis in bone metabolism, and proposed that the combinations of gastrodin with Mg^2+^, ARB2 or artemisinin may have greater potential to anti-osteoporosis. These potential mechanisms and drug combinations are expected to provide new ideas for the study of gastrodin against osteoporosis, and ultimately contribute to the treatment of osteoporosis patients. Thus, this herbal active ingredient should be considered in the treatment of bone diseases and implant osseointegration.

## Figures and Tables

**Figure 1 pharmaceuticals-15-01432-f001:**
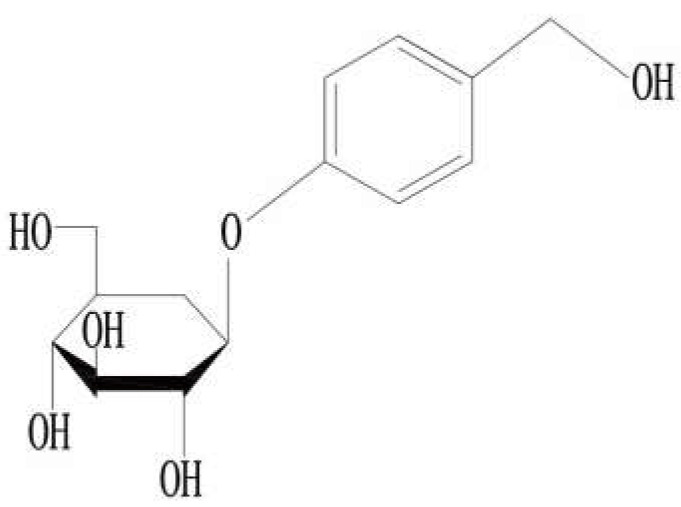
Chemical structures of gastrodin.

**Figure 2 pharmaceuticals-15-01432-f002:**
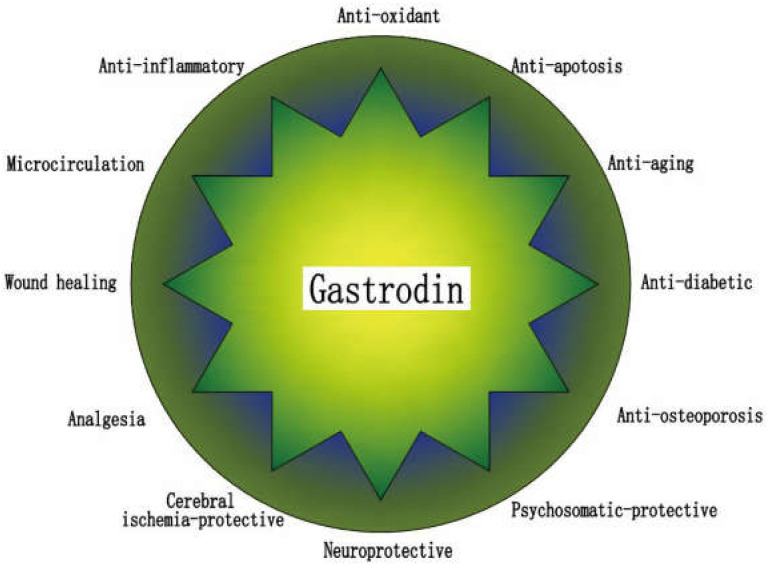
Summary of the beneficial effects of gastrodin.

**Figure 3 pharmaceuticals-15-01432-f003:**
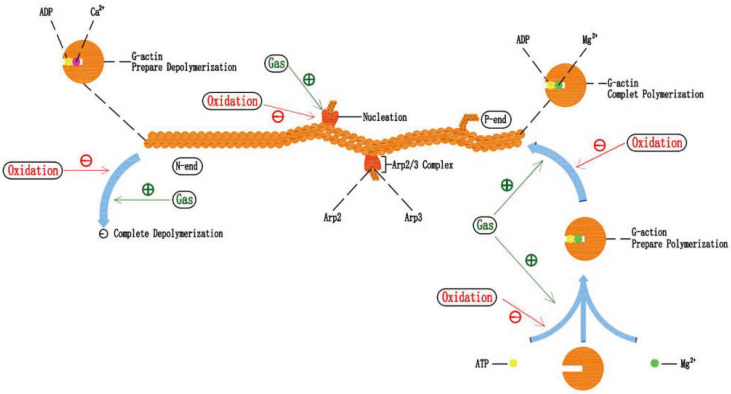
The potential mechanisms of gastrodin protecting actin filament polymerization under oxidative stress. Gastrodin protects the mitochondria to produce ATP, increasing the amount of ATP-bound G-actin, and reduces G-actin oxidation, thus improving the actin filament polymerization rate and reducing the depolymerization rate. At the same time, gastrodin also protects the Arp2/3 complex-induced nucleation process.

**Figure 4 pharmaceuticals-15-01432-f004:**
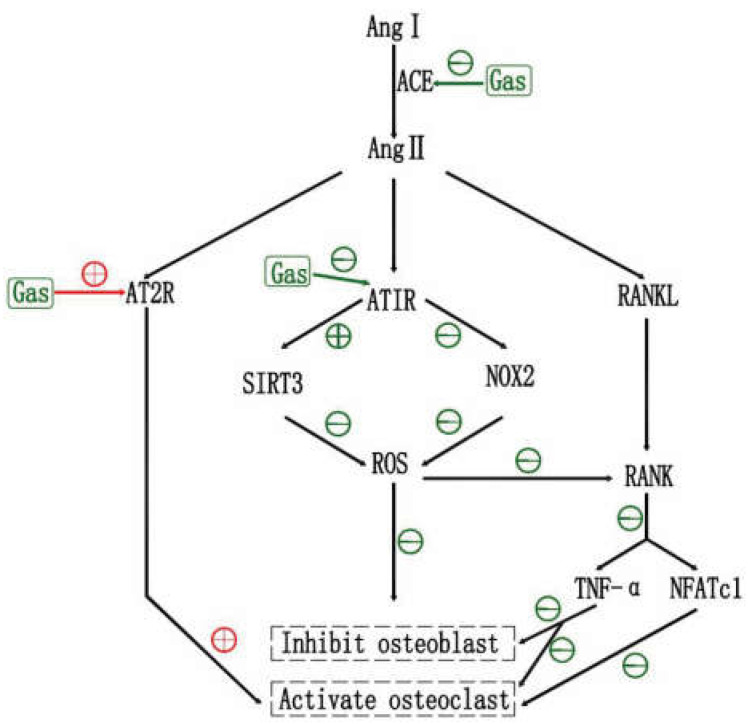
The potential mechanism of gastrodin against osteoporosis via RAS. Gastrodin partially blocks RAS by inhibiting the activation of ACE and AT1R, then decreasing the NOX-2 expression and increasing the SIRT3 expression, eventually reducing ROS content. As a result, RANLKL induced osteoclast differentiation is inhibited and oxidative damage of osteoblast is alleviated. But the increased expression of AT2R induced by gastrodin promotes osteoclast differentiation.

**Figure 5 pharmaceuticals-15-01432-f005:**
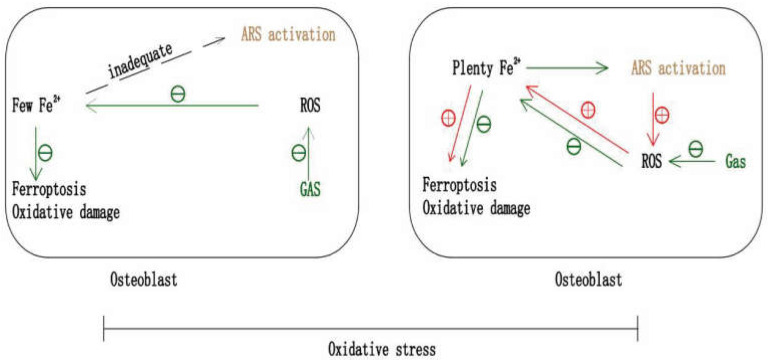
The potential combination of gastrodin and artemisinin for anti-osteoporosis. Gastrodin protects osteoblast and osteoclast from oxidative damage and ferroptosis under oxidative stress. Due to higher intracellular iron content, ARS selectively inhibits osteoclast activity. Therefore, the combination of gastrodin and ARS may have greater therapeutic potential for osteoporosis.

**Figure 6 pharmaceuticals-15-01432-f006:**
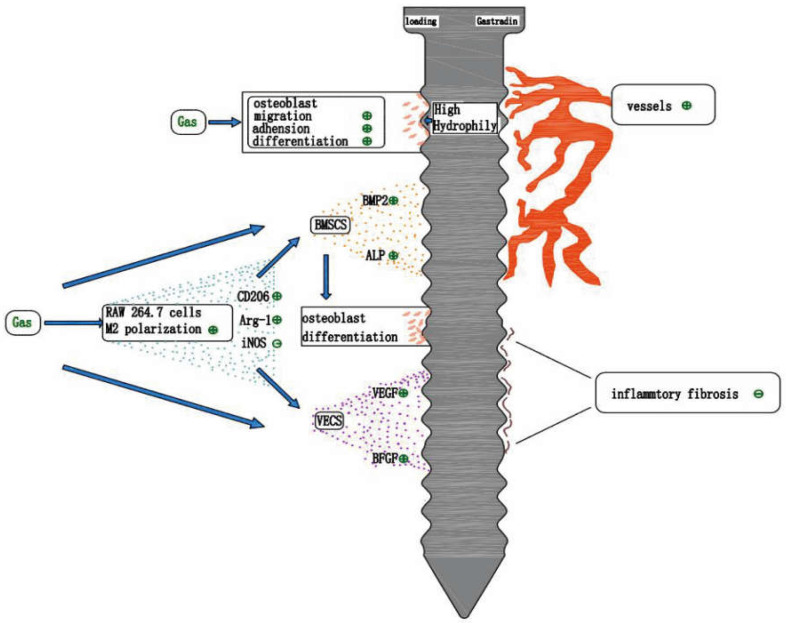
The potential effects of using gastrodin in oral implant surface loading. Gastrodin improves the implant surface hydrophilicity, which is beneficial to cells adhesion. Sustained-release gastrodin promotes the differentiation and migration of osteoblasts. Gastrodin-induced macrophage M2 polarization drives osteogenic differentiation of BMSCs and vascular regeneration, and reduces the production of inflammatory fibers. Finally, the osseointegration process is accelerated.

**Table 1 pharmaceuticals-15-01432-t001:** Summarized effects of gastrodin on bone metabolism in in vivo and in vitro studies.

Model	Type	Inducer	Animal/Cell	Major Findings	RF
Oxidative stress	In vitro	DEX	MC3T3-E1 cells	↑Cell viability, ↑Osteogenic differentiation, ↓ROS, ↑Nrf2/Keapl pathway	[44]
	In vivo	DEX	SD rats	↑BMD, ↑Trabecular microstructure, ↑Skeletal mechanical strength, ↓ROS, ↑Nrf2/Keapl pathway	[67]
	In vitro	H_2_O_2_	HBMMSCs	↑Cell viability, ↑Osteogenic differentiation, ↓Lipogenic differentiation, ↓ROS	[37]
	In vivo	OVX	BALB/c female mice	↑Trabecular microstructure, ↑Mineral apposition rate, ↓MDA, ↑GSH	[37]
	In vivo	High glycolipid + Streptozocin	SD rats	↑Trabecular microstructure, ↓MDA, ↑SOD	[68]
	In vivo	Sodium fluoride	Wistar rats	↑Trabecular microstructure, ↓MDA, ↑CAT	[9]
	In vitro	LPS	hPDLSCs	↑Osteogenic differentiation, ↓ROS, ↓MDA, ↓ LDH, ↑SIRT3	[69]
	In vitro	H_2_O_2_	RAW264.7 cells	↓Osteoclastic differentiation, ↓NFATc1, ↓TRAP, ↓CTR, ↓CTSK	[37]
	In vivo	H_2_O_2_	RAW264.7 cells	↓Osteoclastic differentiation, ↓TRAP, ↓CTX-1	[37]
	In vitro	--	BMMs	↓NFATc1, ↓TRAP, ↓CTSK, ↓DC-STAMP	[51]
Apoptosis	In vitro	Sodium fluoride	MC3T3-E1 cells	↑Cell viability, ↓Caspase 3, ↓Caspase 9, ↓Bax	[9]
	In vivo	Steroid	Wistar rats	↓Osteonecrosis rate, ↓Bax, ↓Caspase 3, ↑Bcl-2	[52]
	In vivo	High glycolipid + Streptozocin	SD rats	↑Trabecular microstructure, ↓Bax, ↑Bcl-2	[68]
	In vitro	DEX	MC3T3-E1 cells	↑Cell viability, ↑Osteogenic differentiation, ↓Caspase 3	[44]
	In vivo	DEX	SD rats	↑BMD,↑Trabecular microstructure, ↑Skeletal mechanical strength, ↑AIF, ↓Caspase 3, ↓Bax, ↑Bcl-2	[67]
	In vitro	LPS	hPDLSCs	↑Osteogenic differentiation, ↓Caspase 3, ↓Caspase 9, ↓Bax, ↑Bcl-2	[69]
	In vitro	1L-1β	Chondrocytes	↑Cell viability, ↓Caspase 3, ↓Bax, ↑Bcl-2	[70]
Inflammation	In vitro	LPS	hPDLSCs	↑Osteogenic differentiation, ↓TNF-α, ↓IL-6	[69]
	In vitro	1L-1β	Chondrocytes	↑Cell viability, ↓TNF-α, ↓IL-6, ↓NF-κB pathway	[70]
	In vivo	OA	SD rats	↑Cartilage structure, ↓OARSI scores, ↓MMP3, ↓TNF-α	[70]
	In vitro	H_2_O_2_	HBMMSCs	↑Cell viability, ↑Osteogenic differentiation, ↓Lipogenic differentiation, ↓RANKL, ↓IL-6	[37]

**Table 2 pharmaceuticals-15-01432-t002:** Summarized effects of gastrodin on organizational engineering and osseointegration in vivo and in vitro studies.

Components	Mode	Type	Tissue/Cell	Major Findings	RF
Organizational engineering	Anti- oxidation	In vitro	HUVECs (H_2_O_2_)	↑Cell viability, ↑Nrf2, ↑HO-1	[17]
	Improvement of hydrophilic	In vitro	RBMSCs	↑Adhesion, ↑Migration	[13]
		In vivo	Femoral condyle defect of rats	↑Osteogenesis, ↑Angiogenesis	[13]
		In vitro	HUVECs; Schwann cells; PC12 cells	↑Adhesion, ↑Migration	[17,143,144]
		In vivo	Subcutaneous pocket of rats	↑Angiogenesis, ↑Nerve regeneration	[17,143]
	Anti- inflammatory	In vitro	RAW264.7 cells, RBMSCs, HUVECs	↑M2 polarization, ↑CD206, ↓Arg-1; ↑BMP-2, ↑ALP; ↑VEGF, ↑BFGF	[13]
		In vivo	Femoral condyle defect of rats	↑Osteogenesis, ↑Angiogenesis	[13]
	pro-vascular regeneration	In vitro	HUVECs	↑Cell viability, ↑Angiogenesis; ↑VEGF	[13]
		In vivo	Femoral condyle defect of rats; Subcutaneous pocket of rats	↑Osteogenesis, ↑Angiogenesis	[13,17,143]
Implant osseointegration	--	In vitro	BMSCs	↑Adhesion on titanium plates, ↑ALP	[51]
	Anti- oxidation	In vivo	Rats (T2DM)	↑Trabecular microstructure around implant, ↓MDA, ↑SOD	[68]

## Data Availability

Data sharing not applicable.

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
