# Peer review of "Mechanism and Prospect of Gastrodin in Osteoporosis, Bone Regeneration, and Osseointegration"

_pharmaceuticals, 2022, doi:10.3390/ph15111432_

Round 1
Reviewer 1 Report
In this manuscript, Yi Li and Fenglan Li reviewed the mechanisms of gastrodin in osteoporosis , bone regeneration and osseointegration, including antioxidant, anti-apoptosis, anti-inflammation, inhibiting bone absorption, increasing material surface hydrophilicity, immunomodulation, pro-vascular regeneration, affecting actin filament polymerization, renin-angiotensin system and ferroptosis. The manuscript summarized a large number of literature relevant for the field, including some very new findings. However, they were not presented in a clear and well-structured manner in this manuscript.
1, The authors reviewed separately in section2 “Oxidative stress menopausal women , elderly patients, and diabetes patients with osteoporosis”, while section1 is “introduction” and section3 is “Gastrodin”. They then reviewed “Gastrodin for OP treatment”in section4, and other three mechanisms of anti-OP in section7-9, after talked about bone regeneration in section5 and implant osseointegration in section6.
2, The authors reviewed the mechanisms or the pharmacological properties of gastrodin which are not on the same level. For example, table 1 summarized effects of gastrodin on bone metabolism in vivo and invitro studies, the left column summarized the model as “oxidative stress” “apoptosis” “inflammation” and “Osteoclast bone resorption”, while the former 3 and the latter one were not on the same level. The same problem is in the text. They reviewed antioxidant, anti-apoptotic and anti-inflammatory effects in 4.1.1-4.1.3, while in 4.2.2, they talked about NFATc1, but not anti-inflammation; in section 5.3, they changed to “immune regulation”. All these brought gaps to the readers and damaged the structure of this article.
I suggest the authors to reorganize the manuscript and they can use the structure as section8 (8.1, 8.2, 8.3) and put OP sections together, then move to section of bone regeneration and section of osseointegration. In every section, the structure can be organized as section 8.
Other comments:
1, Grammatical mistake: Line 115, “ROS aggravates….” , line 125, “ROS contribute to bone…”
2, Figure 2, add references to the pharmacological properties;
3, Table 1 and table 2 , in vivo and in vitro studies had better be put together( in vitro first , then in vivo)
4, Line 679-680, “Artemisinin (ARS), a sesquiterpene lactone compound, is a traditional Chinese medicine isolated from a herb[162].” Artemisinin is not a TCM, but the herb artemisia annua is a TCM.
5, In the subsection “9.3 The potential of gastrodin and ARS in combination against OP”, no reference was cited. Since it is not the main topic of this paper , this subsection can be removed.
Author Response
Response to Reviewer 1 Comments
Point 1: For example, table 1 summarized effects of gastrodin on bone metabolism in vivo and invitro studies, the left column summarized the model as “oxidative stress” “apoptosis” “inflammation” and “Osteoclast bone resorption”, while the former 3 and the latter one were not on the same level. The same problem is in the text. They reviewed antioxidant, anti-apoptotic and anti-inflammatory effects in 4.1.1-4.1.3, while in 4.2.2, they talked about NFATc1, but not anti-inflammation
Response 1: Dear reviewer, thanks for your comments. We make the following explanations and adjustments according to the comments. Section 4 “summarizes the anti osteoporosis effect of gastrodin”, which is divided into two parts: Section 4.1 “protecting bone formation” and Section 4.2 “inhibiting bone absorption”. Section 4.2 is further divided into Section 4.2.1 “Osteoclast inhibition under oxidative stress environment” and Section 4.2.2 “Osteoclast inhibition under normal environment”.
It should be noted that, in Section 4.2.2, the study on gastrodin inhibiting osteoclasts in normal environment [51], the author only pointed out that gastrodin acts through NFATc1, and did not mention the effect of anti-oxidation, anti-inflammatory or anti apoptosis in the conclusion. Through literature review and discussion, we speculate that the mechanism is still antioxidant or to reduce the normal level of ROS. Our discussion of this in the manuscript: “ In a healthy organism, ROS at a normal level participates in the regulation of normal operation of various biological functions. The differentiation of osteoclasts requires the activation of RANKL, and this process needs the involvement of moderate ROS[28]. Compared with other bone cells, osteoclasts need more ROS[92]. Therefore, we speculate that in normal environments, such as this study, gastrodin still inhibits osteoclast differentiation by reducing ROS level. This low level of ROS is insufficient to maintain the activation of RANKL which is required for osteoclast differentiation.”(L298 - L304).
Section 4.2.1 research in oxidative stress environment [37] (L275-L279) and Section 4.2.2 research in normal environment [51] (L288-L292) all belong to the category of antioxidant mechanism of gastrodin. Considering the statements of the original author [51], we have temporarily modified the titles of Section 4.2.1 and Section 4.2.2 based on the difference between oxidative stress environment and normal environment. Specifically, we added "under aggressive stress", deleted "via NFATc1", and added "in normal environment through adverse effect". Finally, "4.2.1 Gastrodin inhibitors osteoblast differentiation under oxidative stress through antioxidant effect" (L268 - L269). Finally,“4.2.2 Gastrodin inhibits osteoclast differentiation in normal environment through antioxidant effect”(L280 - L281)。
Under the guidance of the reviewer's suggestion, we also transferred the text in Section 4.2 to the "Oxidative stress" column in Table 1. So that these terms are at the same level as far as possible. Similarly, we change the "osseointegration" to "Implant osseointegration" in the first column of Table 2.
Point 2: in section 5.3, they changed to “immune regulation”
Response 2: Thanks for your comments. We changed "Involvement of gastrodin in immune regulation" to "Anti-inflammatory effect of gastrodin" (L613). We have adopted "anti-inflammatory" in Abstract (L14), Conclusion (L737) and Table 2.
Point 3: put OP sections together, then move to section of bone regeneration and section of osseointegration.
Response 3: Thanks very much for your suggestion. We have revised the structure of the article, as the reviewer said. Thank you again! Specifically, Section 1 "Introduction". Section 2 "Osteoporosis". Section 3 "Gastrodin". Section 4 "The known mechanism of gastrodin against osteoporosis". Section 5 "Gastrodin and Actin filament". Section 6 "Gastrodin and RAS". Section 7 "Gastrodin and Ferroptosis". Section 8 "Gastrodin and Tissue Engineering". Section 9 "Gastrodin and Implants".
Point 4: In every section, the structure can be organized as section 8
Response 4: Thanks for your comments. We compressed Section 5 into three parts. Since Section 5.1 (L319) only discussed the protection of gastrodin on actin filament at the macro level, and did not mention the polymerization-depolymerization of actin filament, we take this part as the first part. Since Section 5.2 (L343) explained the polymerization-depolymerization process at the micro level for the first time, and discussed the impact of oxidative stress on polymerization-depolymerization at the micro level, we regard this part as the second part.
Point 5: Grammatical mistake: Line 115, “ROS aggravates….” , line 125, “ROS contribute to bone…”
Response 5: Thanks for your comments. We are deeply sorry for such mistakes. We have corrected several errors in this paragraph (L115 - L120). We changed "ROS aggregates...." to "ROS causes...." (L115). We changed "increasing osteolast apicosis..." to "increases osteolast apicosis..." (L118). We changed "causing lipid peroxidation and cellular ferrotropism" to "causes cellular ferrotropism by lipid peroxidation" (L119 - L120). We changed "and inducing osteoblast..." to "and induces osteoblast..." (L120).
Finally,“ROS causes the osteoblast oxidative damage by inhibiting the expression of antioxidant enzymes (nuclear factor erythroid 2-related factor [Nrf2] and hemoxygenase-1 [HO-1]) [44, 45], increases osteoblast apoptosis by inhibiting the phosphoinositide-3-kinase (PI3K)/protein kinase B (AKT) signaling pathway [46], causes cellular ferroptosis by lipid peroxidation [44, 47], and induces osteoblast autophagy through the ROS-AKT-mammalian target of rapamycin axis [48].”(L115 - L120)
We also revised this paragraph (L124-L127). We also revised this paragraph (L124 - L127). We changed "ROS contribute to bone..." to "ROS causes bone loss..." (L125). We changed "damage to the osteopogentor cells and preosteoblasts..." to "damage to the osteoblast..." (L125). We changed "gastrodin protects the osteogenic differentiation of osteoprotector cells and presteoblasts [37, 44]" to "gastrodin protects the osteogenic differentiation of osteoblasts [37, 44]," (L127).
Finally, “Moreover, ROS causes bone loss by causing oxidative damage to the osteoblast and promoting osteoclast differentiation. As an antioxidant drug, gastrodin protects the osteogenic differentiation of osteoblast [37, 44], inhibits osteoclast differentiation [51],”(L124 - L127).
Point 6: Figure 2, add references to the pharmacological properties
Response 6: Thanks for your comments. First, we added references [9, 53-57]. Secondly, considering the preciseness of words, we changed "Figure 2. Summary of the pharmaceutical properties of gastrodin" to "Figure 2. Summary of the beneficial effects of gastrodin" (L160).
Point 7: Table 1 and table 2 , in vivo and in vitro studies had better be put together( in vitro first , then in vivo)
Response 7: Thanks for your comments. Unfortunately, after trying, we found that because a study may involve multiple mechanisms, combining in vitro and in vivo studies will lead to crowding. Therefore, we have not merged the in vivo and in vitro studies in the table for the time being. Please forgive us.
Point 8:“Artemisinin (ARS), a sesquiterpene lactone compound, is a traditional Chinese medicine isolated from a herb[162].” Artemisinin is not a TCM, but the herb artemisia annua is a TCM.
Response 8: Thanks for your comments. We are deeply sorry for such mistakes. We added "ingredient". Finally, "Artemisin (ARS), a sesquiterpene lactone compound, is a traditional Chinese medicine ingredient isolated from a herb [92]." (L518).
In addition, we changed "Gastrodin (Figure 1), also known as red arrow, is the dried tuber of the herb Tian ma," to "Gastrodin (Figure 1) is the extract of dried tuber of the herb Gastrodia elata (Tian ma), " (L130).
Point 9: In the subsection “9.3 The potential of gastrodin and ARS in combination against OP”, no reference was cited. Since it is not the main topic of this paper , this subsection can be removed
Response 9: Thanks for your comments. For this part, we must admit that our previous description is very poor, may leading to misunderstanding. We have made changes and added references. We hope that reviewers could read and consider retaining this part.
Finally, “Based on the possible protective effect of gastrodin on the viability of osteoclasts under oxidative stress [37], the selective inhibitory effect of ARS on osteoclasts [92] and the known osteoprotective effect of gastrodin shown in Table 1, we propose that the combination of gastrodin and ARS may have greater therapeutic potential for OP. This drug combination is characterized by protecting the proliferation and differentiation of pre-osteoblasts [37, 44], inhibiting the differentiation and maturation of pre-osteoclasts [37, 51], and further selectively inhibiting the viability of osteoclasts which can counteract the relatively positive effect of gastrodin on osteoclasts [92, 141-143] (Figure 5). Of course, first of all, it is necessary to verify the protective effect of gastrodin on the viability of osteoclasts through reducing oxidative damage under oxidative stress (the protection on the viability of pre-osteoclasts has been proved [37]). Through this step, we will determine that the combination with ARS, which selectively inhibits osteoclasts, is ingenious and meaningful. Secondly, to verify the inhibition of gastrodin on ferroptosis of osteoblasts and osteoclasts can further indicate the value of this drug combination. Finally, it is important to explore the appropriate dosage combination of the two drugs.” (L532 - L547).
In addition, allow us to explain again. Roughly speaking, the treatment of osteoporosis mainly starts from two points: promoting osteoblasts and inhibiting osteoclasts. Gastrodin protects the differentiation and viability of osteoblasts and pre-osteoblasts under oxidative stress. This is positive. Gastrodin also inhibited the differentiation of osteoclasts. This is also positive. However, gastrodin protects the viability of pre-osteoclasts and mature osteoclasts by reducing oxidative damage and resisting ferroptosis. This is positive for osteoclasts, but relatively negative for anti-osteoporosis. Therefore, artemisinin, which selectively inhibits the activity of osteoclasts by causing oxidative damage and ferroptosis, can ingeniously counteract the positive effect of gastrodin on osteoclasts. The combination of the two drugs further expanded the effect of promoting osteoblasts and inhibiting osteoclasts.
The mechanisms and effects of oxidative damage and ferroptosis involved in this drug combination are shown in Figure 5. This is also the purpose of Figure 5.
In addition, we have also revised the contents of Sections 7.1 and 7.2. The purpose is to clearly point out that gastrodin protects the viability of osteoclasts by reducing oxidative damage and resisting ferroptosis, and that artemisinin inhibits osteoclasts by inducing oxidative damage and ferroptosis. This is conducive to eliciting the ingenuity of drug combination in Section 7.3.
Finally (part of Sections 7.1 ), “Therefore, we suggest that gastrodin may inhibit ferroptosis in osteoblasts through antioxidant effect, thereby protecting osteoblasts viability and alleviating OP.”(L503 -L505)。
“However, it should be noted that although gastrodin can inhibit the differentiation of osteoclasts, gastrodin may also protect osteoclasts viability via inhibiting ferroptosis, especially when osteoclasts are already under oxidative stress. Moreover, It has been proved that gastrodin protects the viability of pre-osteoclasts by reducing intracellular oxidative damage [37]. In the two related studies, gastrodin does not inhibit the proliferation of BMMs [51], but protects the proliferation of RAW264.7 cells under oxidative stress [37]. This enlightens us that counteracting this relatively positive effect of gastrodin on osteoclasts (which is relatively negative for anti-osteoporosis) is conducive to amplifying the anti-osteoporosis effect of gastrodin.”(L506 - L514)。
Finally (part of Sections 7.2 ), “Therefore, Zhang et al. [92] creatively put forward that the high level of intracellular iron makes osteoclast become the potent target of ARS compounds, which can selectively inhibit the differentiation and viability of osteoclasts via inducing intracellular oxidative damage and ferroptosis, but not osteoblasts. In fact, in the current studies, ARS and its derivatives do not affect or have very limited role in differentiation of osteoblasts [92] and BMSCs [136], but can inhibit the differentiation and formation of osteoclasts. [141- 143]” (L524 - L529).

Reviewer 2 Report
Dear authors, the manuscript "Mechanism and prospect of gastrodin inosteoporosis, bone regeneration, and osseointegration" is quite interesting and worth investigation. Please see comments below:
1 - Please double-check enghish grammar and formatting for instance in vivo rather than in vivo.
2 - The manuscript covers a plenty of sub-subjects. It is structured.
3 - Have you considered add information on the scientific trends of gastrodin? Scopus database?
4 - The figures 3 and 4 are very well-drawn.
Regards
Author Response
Response to Reviewer 2 Comments
Point 1: Please double-check enghish grammar and formatting for instance in vivo rather than in vivo.
Response 1: Dear reviewer, thanks for your comments. We are deeply sorry for the grammar and format errors in the manuscript. Thanks for your comments. We changed all "in vivo" and "in virto" to "in vivo" and "in virto". We changed "International Osteoporosis Foundation" to "International Osteoporosis Foundation" (L32). We changed "Tian ma” to "Tian ma" (L130). We changed "- O- β- D -" to "- O- β- D-"(L133)。 We changed "in situ" to "in situ" (L593). We italicized the letters in the subheading.
Point 2: The manuscript covers a plenty of sub-subjects. It is structured.
Response 2: Thanks for your comments. We have adjusted the structure of the manuscript on the original basis. Specifically, Section 1 "Introduction". Section 2 "Osteoporosis". Section 3 "Gastrodin". Section 4 "The known mechanism of gastrodin against osteoporosis". Section 5 "Gastrodin and Actin filament". Section 6 "Gastrodin and RAS". Section 7 "Gastrodin and Ferroptosis". Section 8 "Gastrodin and Tissue Engineering". Section 9 "Gastrodin and Implants".
Point 3: Have you considered add information on the scientific trends of gastrodin? Scopus database?
Response 3: Thank you very much for your suggestions on Scopus database! This is very helpful for us. We didn't know much about the Scopus database before, and we are surprised by the rich information in the database. By searching the Scopus database, we believe that the application of gastrodin in bone metabolism is full of potential. In particular, the application of gastrodin in the functional modification of scaffold materials is very attractive. In this regard, the research conducted by the teams of LimeiLi and MengZheng is very valuable and developing continuously. The content in the database is diverse, and we will use it in more detail and with more focus to stimulate new ideas. Thank you again!
Point 4: The figures 3 and 4 are very well-drawn.
Response 4: We thank you very much for your comments! Your affirmation is our greatest encouragement and great driving force for our progress. We will continue to work hard. thank you!

Reviewer 3 Report
1) Abstract. Gastrodin, a traditional Chinese medicine ingredient, is mostly used to treat vascular and neurological diseases, owing to its antioxidant, anti-inflammatory and and other beneficial properties. However, recently, an increasing number of studies have been conducted on the inhibition of bone loss by gastrodin. Therefore, this paper reviewed the relationship between bone loss and oxidative stress in populations with a high prevalence of osteoporosis, such as menopausal women, elderly patients, and diabetes patients, as well as the the effects and the mechanisms of gastrodin on osteoporosis, bone regeneration and osseointegration. Specifically, the mechanisms involved include antioxidant, anti-apoptosis, anti-inflammation, inhibiting bone absorption, increasing material surface hydrophilicity, immunomodulation, and pro-vascular regeneration. We also hypothesized that the use of gastrodin in oral implant surface loading has great potential. In addition, We also speculated on the potential mechanisms of gastrodin against osteoporosis by affecting actin filament polymerization, renin-angiotensin system and ferroptosis, and proposed that the combinations of gastrodin with Mg2+, angiotensin type 2 receptor blockers or artemisinin may have greater potential to anti-osteoporosis. We hope to provide a reference for more extensive and in-depth research on the application of gastrodin in the treatment of osteoporosis and implant osseointegration in the future. The abstract is quite rumbling and difficult to read. Please, divide the paragraph in different sections such as background, aim and discussions
2) Introduction. L 59-62. Moreover, we also hypothesizedthepotential59 mechanisms of gastrodin on actin filament polymerization, RAS and anti-ferroptosisinbone metabolism, and proposed that the combinations of gastrodinwithMg2+60 , angiotensin type 2 receptor blockers or artemisinin may have greater potentialto anti-osteoporosis. Could you please insert this part in the conclusions?
3) 2. Oxidative stress in menopausal women, elderly patients, and diabetes patientswith osteoporosis.
L64-70. OP is characterized by very poor bone quality due to the loss of bonemineral density (BMD), microarchitecture, and mechanical strength. It is a major causeof skeletal fragility and dysfunction in postmenopausal women and elderlypatients[18]. Approximately one-third of postmenopausal women aged >50 years are affectedbyOP. As the population ages, the incidence of osteoporotic fractures is increasing[19]. Growing evidence has suggested that diabetes is strongly associatedwiththe development of OP[19]. Op is increased also in several systemic diseases. In order to discuss the previously described points, important references are needed to be added, such as:
A- Evaluation of Dexamethasone-Induced Osteoporosis In Vivo Using Zebrafish Scales. Pharmaceuticals 2021, 14, 536. https://doi.org/10.3390/ph14060536
B- Correlation between bone quality and microvascular damage in systemic sclerosis patients. Rheumatology (Oxford). 2018;57(9):1548-1554. doi:10.1093/rheumatology/key130
4) 4. Gastrodin for osteoporosis treatment L163-166. An increasing number of studies have demonstrated the potential of gastrodinin the treatment of OP. Through its antioxidant, anti-apoptotic, and anti-inflammatory properties, gastrodin promotes the viability and osteogenic differentiationof osteoprogenitor cells, preosteoblasts, and periodontal stemcells and inhibits osteoclast. Please, underline in the text the most important data to support the sentences and results of most important studies.
5) 10. Conclusions L708-721. Whether it is bone loss induced by estrogen deficiency, age, or hyperglycemiaor bone loss due to osteonecrosis or periodontitis, gastrodin has shown goodtherapeutic effects in vivo and in vitro against OP. Additionally, it has shown beneficial effectson tissue engineering, bone healing, and implant osseointegration. Its mechanismsof action include its antioxidant effect, anti-inflammatory effect, anti-osteoclast apoptosis, inhibition of osteoclast differentiation, inducing high hydrophilicity inthematerial surface, immunomodulation, and pro-vascular regeneration. However, the mechanisms by which gastrodin affects bone, especially osseointegration of biomaterials, requires further detailed research, as there are relatively few relevant clinical trial studies. Furthermore, given the characteristics of gastrodin and the many advantagesof promoting bone healing by loading gastrodin into the scaffold, the use of gastrodinin the surface loading of oral implants may have great potential. However, thereisa paucity of research on osseointegration of oral implants and its clinical applicationsin this area. Please improve this paragraph and underline the novelty of the study and the possible clinical implications.
Author Response
Response to Reviewer 3 Comments
Point 1: The abstract is quite rumbling and difficult to read. Please, divide the paragraph in different sections such as background, aim and discussions
Response 1: Dear reviewer, thanks for your comments. To make the summary more organized, we rewrote the summary section (L8-L24). The first half (L8-L14) mainly describes the traditional and new application background of gastrodin, as well as the research background of the known effects and mechanisms of gastrodin on osteoporosis and bone healing. The second half (L15-L22) mainly discusses our prospects for the role of gastrodin in bone, including the potential anti-osteoporosis mechanisms of gastrodin, more powerful drug combinations, and the prospects of gastrodin in implant osteointegration. The end part (L22-L24) mainly indicates the purpose of the review.
Finally, “Gastrodin, a traditional Chinese medicine ingredient, is widely used to treat vascular and neurological diseases. However, recently, an increasing number of studies have shown that gastrodin has anti-osteoporosis effects, and Its mechanisms of action include its antioxidant effect, anti-inflammatory effect and anti-apoptosis effect. In addition, gastrodin has many unique advantages in promoting bone healing in tissue engineering, such as inducing high hydrophilicity in the material surface, anti-inflammatory effect, and pro-vascular regeneration. Therefore, this paper summarized the effects and mechanisms of gastrodin on osteoporosis and bone regeneration in the current researches. And we proposed a assumption that the use of gastrodin in the surface loading of oral implants may greatly promote the osseointegration of implants and increase the success rate of implants. In addition, we speculated on the potential mechanisms of gastrodin against osteoporosis by affecting actin filament polymerization, RAS and ferroptosis, and proposed that the potential combination of gastrodin with Mg2+, angiotensin type 2 receptor blocker or artemisinin may greatly inhibit osteoporosis. The purpose of this review is to provide a reference for more in-depth research and application of gastrodin in the treatment of osteoporosis and implant osseointegration in the future.” (L8 - L24).
Point 2: Introduction. L 59-62. Moreover, we also hypothesizedthepotential59 mechanisms of gastrodin on actin filament polymerization, RAS and anti-ferroptosisinbone metabolism, and proposed that the combinations of gastrodinwithMg2+60 , angiotensin type 2 receptor blockers or artemisinin may have greater potentialto anti-osteoporosis. Could you please insert this part in the conclusions?
Response 2: Thanks for your comments. We modified the above content and inserted it into the “Conclusion”. (L733-L736)
Finally (part of Conclusion), “In addition, based on the existing research, we also hypothesized the potential mechanisms of gastrodin affecting on actin filament polymerization, RAS and ferroptosis in bone metabolism, and proposed that the combinations of gastrodin with Mg2+, ARB2 or artemisinin may have greater potential to anti-osteoporosis. These potential mechanisms and drug combinations are expected to provide new ideas for the study of gastrodin against osteoporosis, and ultimately contribute to the treatment of osteoporosis patients. Thus, this herbal active ingredient should be considered in the treatment of bone diseases and implant osseointegration.” (L743 - L750).
Point 3: Op is increased also in several systemic diseases. In order to discuss the previously described points, important references are needed to be added, such as:
A- Evaluation of Dexamethasone-Induced Osteoporosis In Vivo Using Zebrafish Scales. Pharmaceuticals 2021, 14, 536. https://doi.org/10.3390/ph14060536
B- Correlation between bone quality and microvascular damage in systemic sclerosis patients. Rheumatology (Oxford). 2018;57(9):1548-1554. doi:10.1093/rheumatology/key130
Response 3: Thanks for your comments. For the rigorism of the manuscript, we have added two important references ([21], [22]) here (L69-L71) and revised the content (L69-L71).
Finally, “Growing evidence has suggested that systematic diseases such as diabetes and rheumatic inflammatory diseases are strongly associated with the development of OP [20, 21]. In addition, glucocorticoids tend to induce iatrogenic osteoporosis [22].” (L69-L71).
[21] Correlation between bone quality and microvascular damage in systemic sclerosis patients. Rheumatology (Oxford). 2018;57(9):1548-1554. doi:10.1093/rheumatology/key130
[22] Evaluation of Dexamethasone-Induced Osteoporosis In Vivo Using Zebrafish Scales. Pharmaceuticals 2021, 14, 536. https://doi.org/10.3390/ph14060536
Point 4: Gastrodin for osteoporosis treatment L163-166. An increasing number of studies have demonstrated the potential of gastrodinin the treatment of OP. Through its antioxidant, anti-apoptotic, and anti-inflammatory properties, gastrodin promotes the viability and osteogenic differentiationof osteoprogenitor cells, preosteoblasts, and periodontal stemcells and inhibits osteoclast. Please, underline in the text the most important data to support the sentences and results of most important studies.
Response 4: Thanks for your comments. Table 1, or Section 4.1.1, 4.1.2, 4.1.3, 4.2.1, and 4.2.2 can support these important statements in the manuscript (L163-L166). As there are many contents, we didn't draw them with lines. Please forgive us. But we have made a simple enumeration below. This part of the sentence is equivalent to a summary description of the anti osteoporosis effect of gastrodin, which is discussed in detail below the manuscript.
Antioxidant: [9], [37], [44], [73], [82],
Anti-apoptotic: [9], [44], [52], [73], [79], [82], [85]
Anti-inflammatory: [37], [79], [85]
Promoting osteoblast: [9], [37], [44], [79], [85]
Inhibiting osteoclast: [37], [51]
[9]Zheng B, Shi C, Muhammed F K, et al. Gastrodin alleviates bone damage by modulating protein expression and tissue redox state[J]. FEBS open bio, 2020,10(11):2404-2416.
[37]Huang Q, Shi J, Gao B, et al. Gastrodin: an ancient Chinese herbal medicine as a source for anti-osteoporosis agents via reducing reactive oxygen species[J]. Bone, 2015,73:132-144.
[44]Liu S, Fang T, Yang L, et al. Gastrodin protects MC3T3-E1 osteoblasts from dexamethasone-induced cellular dysfunction and promotes bone formation via induction of the NRF2 signaling pathway[J]. International journal of molecular medicine, 2018,41(4):2059-2069.
[51] Zhou F, Shen Y, Liu B, et al. Gastrodin inhibits osteoclastogenesis via down-regulating the NFATc1 signaling pathway and stimulates osseointegration in vitro[J]. Biochem Biophys Res Commun, 2017,484(4):820-826.
[52] Zheng H, Yang E, Peng H, et al. Gastrodin prevents steroid-induced osteonecrosis of the femoral head in rats by anti-apoptosis[J]. Chin Med J (Engl), 2014,127(22):3926-3931.
[73] Liu S, Zhou L, Yang L, et al. Gastrodin alleviates glucocorticoid induced osteoporosis in rats via activating the Nrf2 signaling pathways[J]. Oncotarget, 2018,9(14):11528-11540.
[79] Feng Q. Gastrodin attenuates lipopolysaccharide-induced inflammation and oxidative stress, and promotes the osteogenic differentiation of human periodontal ligament stem cells through enhancing sirtuin3 expression[J]. Exp Ther Med, 2022,23(4):296.
[82] Zhang Jingyi L F W H. Preliminary study on the effect of gastrodin on bone tissue around implants in type 2 diabetes rats[J]. Chinese Journal of Stomatology, 2022,57(09):938-945.
[85] Chen J, Gu Y T, Xie J J, et al. Gastrodin reduces IL-1beta-induced apoptosis, inflammation, and matrix catabolism in osteoarthritis chondrocytes and attenuates rat cartilage degeneration in vivo[J]. Biomed Pharmacother, 2018,97:642-651.
Point 5: Conclusions L708-721. Please improve this paragraph and underline the novelty of the study and the possible clinical implications.
Response 5: Thanks for pointing out the defects in the conclusion part of the manuscript. We modified the conclusion (L732-L750) to highlight the novelty and clinical significance of this manuscript. In the previous paragraph (L732-L742), we simplified the description of language, but emphasized the potential and clinical significance of gastrodin in promoting the osseointegration of oral implants. And by showing the lack of relevant research, we highlight the novelty of this idea. In the latter paragraph (L743-L748), we first added the content (L57-L61) to highlight the novelty of the manuscript and the possible clinical value. Finally, we point out that scholars should pay attention to the value of gastrodin in bone metabolism (L749 - L750).
Finally, “Gastrodin is found to have anti-osteoporosis effects. Its mechanisms of action include its antioxidant effect, anti-inflammatory effect, anti-osteoblast apoptosis, inhibition of osteoclast differentiation. Furthermore, given the many unique advantages of gastrodin in promoting bone healing in tissue engineering, such as inducing high hydrophilicity in the material surface, anti-inflammatory effect, and pro-vascular regeneration, we suggest that the use of gastrodin in the surface loading of oral implants may have great potential and effectively promote implant osseointegration. This assumption may provide inspiration for improving the success rate of patients' implants in clinical practice. However, there is a paucity of research on gastrodin as an auxiliary drug for oral implants osseointegration. Particularly, the research on gastrodin involved in local sustained release systems on the surface of oral implants is completely blank.
In addition, based on the existing research, we also hypothesized the potential mechanisms of gastrodin affecting on actin filament polymerization, RAS and ferroptosis in bone metabolism, and proposed that the combinations of gastrodin with Mg2+, ARB2 or artemisinin may have greater potential to anti-osteoporosis. These potential mechanisms and drug combinations are expected to provide new ideas for the study of gastrodin against osteoporosis, and ultimately contribute to the treatment of osteoporosis patients. Thus, this herbal active ingredient should be considered in the treatment of bone diseases and implant osseointegration.” (L732-L750).

Round 2
Reviewer 1 Report
I have no new comments.